# Nucleotide context models outperform protein language models for predicting antibody affinity maturation

Mackenzie M. Johnson[1,☯], Kevin Sung[1,☯], Hugh K. Haddox[1], Ashni A. Vora[2], Tatsuya Araki[2], Gabriel D. Victora[2,3], Yun S. Song[4,5], Julia Fukuyama[6], Frederick A. Matsen IV[1,3,7,8]*

**1** Computational Biology Program, Fred Hutchinson Cancer Center, Seattle, Washington, United States of America, **2** Laboratory of Lymphocyte Dynamics, The Rockefeller University, New York, New York, United States of America, **3** Howard Hughes Medical Institute, Chevy Chase, Maryland, United States of America, **4** Computer Science Division, University of California, Berkeley, Berkeley, California, United States of America, **5** Department of Statistics, University of California, Berkeley, Berkeley, California, United States of America, **6** Department of Statistics, Indiana University Bloomington, Bloomington, Indiana, United States of America, **7** Department of Genome Sciences, University of Washington, Seattle, Washington, United States of America, **8** Department of Statistics, University of Washington, Seattle, Washington, United States of America

☯ These authors also contributed equally to this work.
* matsen@fredhutch.org

## Abstract

Antibodies play a crucial role in adaptive immunity. They develop as B cell receptors (BCRs): membrane-bound forms of antibodies that are expressed on the surfaces of B cells. BCRs are refined through affinity maturation, a process of somatic hypermutation (SHM) and natural selection, to improve binding to an antigen. Computational models of affinity maturation have developed from two main perspectives: molecular evolution and language modeling. The molecular evolution perspective focuses on nucleotide sequence context to describe mutation and selection; the language modeling perspective involves learning patterns from large data sets of protein sequences. In this paper, we compared models from both perspectives on their ability to predict the course of antibody affinity maturation along phylogenetic trees of BCR sequences. This included models of SHM, models of SHM combined with an estimate of selection, and protein language models. We evaluated these models for large human BCR repertoire data sets, as well as an antigen-specific mouse experiment with a pre-rearranged cognate naive antibody. We demonstrated that precise modeling of SHM, which requires the nucleotide context, provides a substantial amount of predictive power for predicting the course of affinity maturation. Notably, a simple nucleotide-based convolutional neural network modeling SHM outperformed state-of-the-art protein language models, including one trained exclusively on antibody sequences. Furthermore, incorporating estimates of selection based on a custom deep mutational scanning experiment brought only modest improvement in predictive power. To support further research, we introduce EPAM

**Data availability statement:** Our models, inference code, and analysis scripts are publicly available on GitHub at https://github.com/matsengrp/epam. All processed data files are publicly available on Zenodo at https://doi.org/10.5281/zenodo.17353498.

**Funding:** This work was supported by National Institutes of Health grants R01-AI146028 (PI Matsen), R56-HG013117 (PI Song) and R01-HG013117 (PI Song). Scientific Computing Infrastructure at Fred Hutch was funded by ORIP grant S10OD028685. Frederick Matsen is an investigator of the Howard Hughes Medical Institute. The funders had no role in study design, data collection and analysis, decision to publish, or preparation of the manuscript.

**Competing interests:** I have read the journal's policy and the authors of this manuscript have the following competing interests: G.D.V. is an advisor for and holds stock of the Vaccine Company. T.A. is currently an employee of Pfizer Inc.

(Evaluating Predictions of Affinity Maturation), a benchmarking framework to integrate evolutionary principles with advances in language modeling, offering a road map for understanding antibody evolution and improving predictive models.

## Author summary

In the immune response to infection or vaccination, antibodies get refined through cycles of mutation and selection. Computational models seek to predict changes in the antibody sequence during this process. We present an analysis of the predictive performance of various models, evaluated on datasets of antibody sequences from humans and mice. We find that more accurate models have a more precise description of the mutation aspect, which requires knowledge of the nucleotide sequence context. In contrast, sophisticated language models trained on large data sets of protein sequences consider only the amino acid context and are less performant. Our findings suggest that antibody language models could improve their predictive power by incorporating the nucleotide context. For this research, we developed a software framework to generate predictions and evaluate performance metrics in a standard way for models with different approaches.

## Introduction

Antibodies are an essential component of the adaptive immune system. B cell receptors (BCRs) are membrane-bound forms of antibodies that are expressed on the surfaces of B cells. These BCRs are created by V(D)J recombination during B cell development, where V, D, and J gene segments are randomly recombined to create the unique antigen-binding regions. Upon antigen stimulation, BCRs undergo *affinity maturation* in the germinal center, which involves cycles of somatic hypermutation (SHM) of the BCR gene sequence followed by natural selection for stronger BCR-antigen binding.

The ensemble of BCR sequences that code for these antibodies within an individual is called the *repertoire*. BCR repertoires capture the immunological history of an individual, including the diversity of BCRs that have been generated in response to exposures to various antigens. High-throughput sequencing has enabled the generation of large-scale BCR repertoire data sets, many of which have been compiled into publicly available databases [1–5]. These data have been used to study the evolution of antibodies in various contexts. Despite stochasticity in the B cell response [6,7], factors that drive outcomes of affinity maturation have been identified (e.g. [8–11]).

Broadly speaking, efforts to learn from large-scale BCR repertoire data have taken the perspective of either molecular evolution (ME) or language modeling (LM). The ME perspective acknowledges that BCRs are nucleotide sequences and investigates processes of SHM and natural selection. For SHM, the nucleotide sequence context is used to train models to learn per-site mutation rates for antibody sequences.

These SHM rates differ between sites, because the enzymes involved preferentially target specific sequence motifs. Identification of mutation hotspots (e.g. [12]) paved the way for probabilistic models of SHM such as the classic S5F model [13] and other more recent models [8,14–16]. For selection on top of this background of SHM, researchers have developed aggregate codon-based models that apply on the entire BCR locus at once [17,18], while others [19] inferred per-site patterns of natural selection per V gene.

In contrast, the LM perspective views BCRs as protein sequences only, which are trained to fit large volumes of curated sequence data [5]. There are now many antibody language models [20–27]. These are most commonly masked language models (MLMs) that are trained to predict the "missing" amino acid at a site given the rest of the antibody sequence. They do not explicitly model V(D)J recombination, SHM, or selection. Rather, they learn patterns that implicitly contain all these effects, which are difficult to disentangle afterward [28].

Although ME and LM represent different perspectives, they are fundamentally describing the same process. In order to understand the ability of current models from either perspective to predict the course of affinity maturation, a unified framework for evaluation is needed.

In this paper, we compared and combined the ME and LM perspectives on antibody affinity maturation by recasting all models into a single framework for evaluation. Specifically, these are models of SHM, models of both SHM and selection, and LMs. Our framework handles models with different inputs (i.e. nucleotide or amino acid sequence) and outputs (e.g. probabilities or mutation rates), and standardizes their predictions for direct comparison. This automated and extensible pipeline is called EPAM for Evaluating Predictions of Affinity Maturation (https://github.com/matsengrp/epam/).

With EPAM, we evaluated the performance for a variety of ME and LM models on large human repertoire data sets, as well as a mouse data set generated from repeated trials of a naive sequence responding to a specific antigen. In both antigen-averaged human repertoire data and antigen-specific mouse repertoires, we found that an accurate characterization of SHM provides considerably more predictive power than the means of modeling amino acid preferences during selection considered here. As a surprising corollary, we found that a simple convolutional neural network (CNN) with a few thousand parameters trained on antibody nucleotide sequences could significantly outperform amino-acid-based language models with many million parameters for prediction of affinity maturation. These findings suggest that LM models could improve their predictive power by making use of the nucleotide context to better incorporate the principles of molecular evolution. Through this work, we hope to chart a course for future research that will bring the best of both perspectives to bear on the problem of understanding antibody evolution.

## Results

### Overview of data preparation and EPAM model evaluation

Our objective was to assess the performance of various models on predicting amino acid substitutions over the course of affinity maturation in BCR repertoires. To meet this objective, we devised EPAM, a unified framework for evaluating models of affinity maturation. Following Spisak et al. [14], data preparation for EPAM involved identifying clonal families from BCR repertoires and reconstructing the phylogenetic tree with ancestral sequences (Fig 1A). The branches of these trees describe evolution between parent-child pairs (PCPs) of sequences. We processed several human repertoire data sets of BCR heavy chain sequences: Ford et al. [29], Rodriguez et al. [30], Jaffe et al. [31], and Tang et al. [15]. The number of PCPs in each data set ranged from 2,407 (Ford et al.) to 438,302 (Tang et al.). The median number of substitutions per PCP ranged from 2–4. It should be noted that Jaffe et al. and part of Tang et al. are in the training data set for the AbLang2 [28] model, while Jaffe et al. is the training data set for the Thrifty-prod model (see Overview of models). We found consistent results with Tang et al., Rodriguez et al., and Ford et al.—the latter two data sets represent truly out-of-sample evaluations, suggesting that biases due to Tang et al. sequences shared by training and evaluation had limited effect.

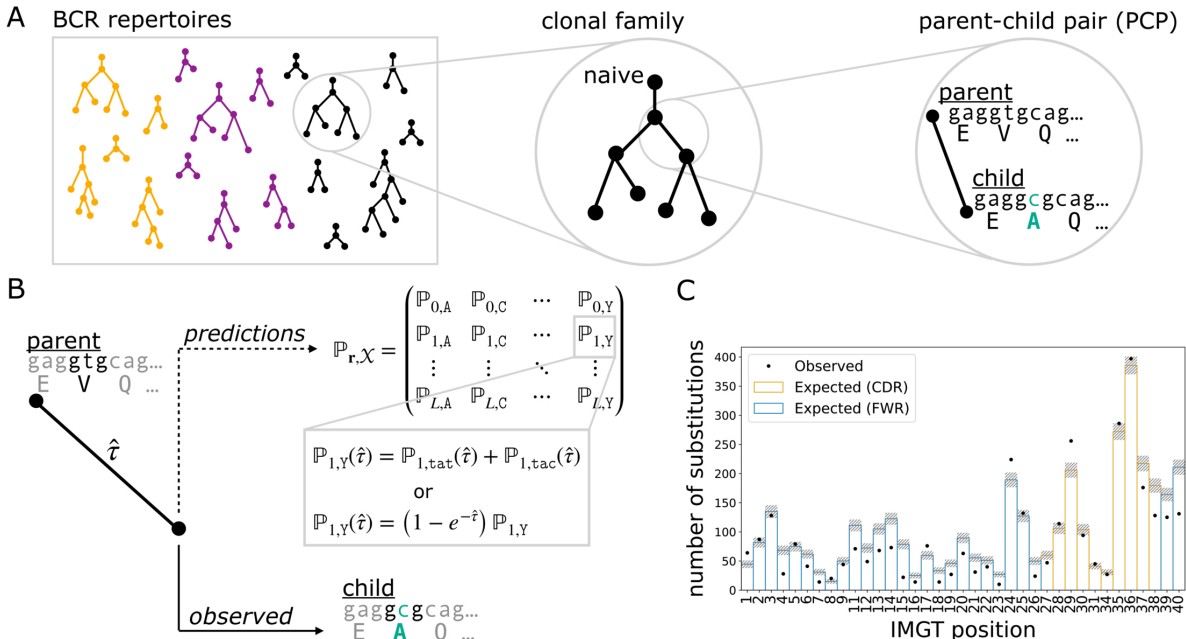

**Fig 1. Data preparation (A) and EPAM framework for assessing model performance (B,C).** (A) BCR DNA sequences from repertoires of multiple individuals (colors) were clustered into clonal families, a phylogenetic tree was reconstructed for each lineage, and the branches provided as parent-child pairs (PCPs) for analysis. PCPs containing the inferred naive sequence were excluded to mitigate the impact from errors in germline inference. (B) The parent sequence and branch length were used to produce model-specific predictions for amino acid probabilities at every site. For nucleotide sequence models, the amino acid probability is the sum over all possible codons that encode the same amino acid (e.g., `tat` and `tac` for tyrosine). We found the optimal branch length ($\hat{\tau}$): the evolutionary distance between parent and child that maximizes the likelihood of the model. For molecular evolution models, $\hat{\tau}$ scales nucleotide mutation rates; for language models, $\hat{\tau}$ scales amino acid probabilities. (C) The amino acid probabilities were used to compute performance metrics for how well a model predicts the course of affinity maturation seen in PCPs. One such metric is the overlap between the observed (dots) and expected (bars) counts distributions of site substitutions.

We also analyzed data from the Replay [32] experiment, where affinity maturation in mice was engineered to play out from the same naive sequence and stimulated by the same antigen. The Replay project provided data under well-controlled conditions and tailored models for SHM and mutation fitness, in contrast to the diversity in human repertoire data and the general purpose models used to analyze them. There were about 5,000 PCPs each in heavy chain sequences and light chain sequences, with a median of one substitution per PCP.

Table A in S1 Text provides a summary of all data sets.

Model evaluation in EPAM was performed on PCPs. Given a model, EPAM generates predictions for a PCP in the form of probabilities for all amino acid outcomes at every site of the parent sequence (Fig 1B). These probabilities depend not only on the parent sequence context, but also on the branch length that reflects the degree of evolution. We computed the optimal branch length that maximized the likelihood to observe the child sequence (details provided in Methods). The predictions were then compared against the substitutions observed in the PCP to see how closely they aligned. Comparisons over all PCPs in a data set were summarized with performance metrics (Fig 1C).

**Performance metrics.** We assessed the performance of a model on two tasks: first, predicting the sites of amino acid substitution, and second, predicting the specific amino acid outcome at substituted sites. These were evaluated over all PCPs in a data set. (The term *substitution* will always refer to a change in amino acid—due to nonsynonymous nucleotide changes in the associated codon.) For each prediction task we had two metrics: one for how often the observed substitutions were also the most probable outcomes according to the model, and another for how well model probabilities were calibrated to data.

To evaluate the ability of models to predict sites of amino acid substitution, we compared the observed and expected counts distributions for the number of substitutions over sequence positions for data and model (Fig 1C) as follows. First we summed the probabilities for amino acid outcomes that differ from the parent sequence at each site of a PCP. We call this the site substitution probability (SSP). To obtain the expected count of nonsynonymous substitutions, we summed the SSPs across all PCPs at each IMGT position [33]. To obtain the observed counts, we tallied the number of substitutions across all PCPs of a data set at each IMGT position. The area of a counts distribution is the sum of the number of substitutions over all sites. Therefore, the areas of the observed and expected distributions are respectively the total number of substitutions observed in the data set and the total expected by the model.

The agreement of the two counts distributions was quantified by the *overlap* metric. This metric is the ratio of the area of intersection to their average area, namely

$$\frac{\sum_i \min(O_i, E_i)}{\sum_i (O_i + E_i)/2},$$

where $i$ runs over the site positions, and $O_i$ and $E_i$ are respectively the observed and expected values at site $i$. An overlap of 1 indicates perfect agreement.

The other metric for sites of substitution was *R-precision*, which quantifies how often the observed substitution sites in a PCP align with the sites the model predicted were most likely to change. If there are $k$ observed substitutions in a PCP, the R-precision is the fraction of those sites being among the top $k$ SSPs predicted by the model. In cases where the $k^{th}$ highest SSP can be multiple sites, the site is randomly selected. We reported the average R-precision over all PCPs in a data set.

To evaluate modeling of amino acid outcomes, we computed the conditional substitution probabilities (CSPs) of each site—the probability of each amino acid outcome given that a substitution occurred. More precisely, given parent nucleotide sequence $S$ and amino acid $\mathcal{X}_{S,r}$ encoded at codon site $r$, we calculated $p\left(\mathcal{X}'|S, \mathcal{X}_{S,r} \neq \mathcal{X}'\right)$, the probability for amino acid outcome $\mathcal{X}'$, conditioned on the parent sequence context and a substitution at $r$. The CSPs are closely related to the predictions of non-germline-encoded sites described in Olsen et al. [28]. Both are estimates of amino acid change at a site relative to an ancestor. However, our CSPs describe outcomes that may occur given the parent sequence, while Olsen et al. [28] is considering how likely it was for the substitution to have occurred given the observed sequence.

We aggregated substitution events across all PCPs in a data set. All substitution events are counted equally, regardless of the PCP in which they occur. The fraction of substitution events where the amino acid outcome is also predicted to have the highest CSP is the *substitution accuracy* of the model. Another metric for predicting the amino acid identity given a substitution is *CSP perplexity*, which we computed for a model on a data set as follows,

$$\left(\prod_{i=1}^{N} \frac{1}{p\left(\mathcal{X}'|S_i, \mathcal{X}_{S_i, r_i} \neq \mathcal{X}'\right)}\right)^{\frac{1}{N}},$$

where the index $i$ enumerates all substitution events that occurred in the data set. A lower perplexity score indicates a better model fit. A hypothetically perfect score is 1, however, we would not expect that to be achieved for any model because affinity maturation is stochastic and typically multiple amino acids are suitable choices at a given site. In comparison to our substitution accuracy metric, CSP perplexity accounts for the probabilities of amino acid predictions rather than just the amino acid that is top ranked.

We present results in the main text for the largest human data set, Tang et al., and the Replay experiment. As stated above, the Tang et al. data set is partially included in the training data for the AbLang2 model. While Rodriguez et al. and Ford et al. do not overlap with any training data set, they have considerably fewer sequences. The results with the Rodriguez et al. (Table B in S1 Text) and Ford et al. (Table C in S1 Text) data sets were qualitatively consistent with Tang

et al. (Table D in S1 Text) and indicates that biases from overlap of training and evaluation data had little impact. In particular, the performance of AbLang2 on Tang et al. was not strikingly different from its performance on Rodriguez et al. and Ford et al.. This gives confidence that our conclusions based on Tang et al. are robust. In contrast, Jaffe et al. makes up a larger portion of the training data for AbLang2 and Thrifty-prod, and we found much better substitution accuracy and CSP perplexity for those two models on that data set (Table E in S1 Text).

## Overview of models

Affinity maturation proceeds through a combination of SHM and natural selection. Some of the models considered cover effects from both these biological processes, while others describe only one of them. We also used certain models in combination in order to account for both processes. How predictions were computed depended on whether the model used the nucleotide or amino acid sequence context. Accordingly, model evaluations were performed under one of three modeling frameworks: "nucleotide" (NT), "nucleotide and amino acid" (NT-AA), and "amino acid" (AA). For any of these frameworks, however, we were always evaluating the ability of a model to predict amino acid substitutions.

We briefly describe these frameworks and the associated models. Classifications for the models used are further listed in Table F in S1 Text. The probability calculations are detailed in the Methods section. The set of models implemented here is not exhaustive, but is representative of the types of models that have been used in the literature.

**NT framework.** Somatic hypermutation is a complex process with highly nonuniform mutation probabilities that are associated with nucleotide motifs [34]. To evaluate models of SHM, we transformed nucleotide-based SHM models into codon models that gave the probability of amino-acid-level mutations arising between PCPs. We considered two SHM models for human data under this NT framework: S5F [13] and ThriftyHumV0.2-59 [16] (which will be denoted as Thrifty-SHM). Both models were trained on data sets where selection pressures were absent in order to isolate the neutral mutation process of antibody evolution. S5F is a classic model of SHM that predicts the probability of a mutation at a given site based on the 5-mer nucleotide context; the model parameters were computed from four-fold synonymous sites in *productive* BCR sequences (sequences that code for functional proteins). Thrifty-SHM considers a wider 13-mer context treated as an arrangement of semi-overlapping 3-mers, and used a CNN on 3-mer embeddings to determine the mutation rate at a site; the training data were *non-productive* BCR sequences, which arise due to premature stop codons or the V and J gene segments being out of frame with each other. There are 5,931 parameters in Thrifty-SHM, which is about twice the number of parameters in S5F (3,072), but orders of magnitude less than protein sequence language models like AbLang2 (45 million) [28] and ESM-1v (650 million) [35].

The Replay project inferred a tailored model that predicted SHM probabilities at each nucleotide site based on the naive sequence context using a passenger mouse system [32]. We will call this model ReplaySHM.

We also considered a model that uses the nucleotide context to describe SHM and selection effects. This model, Thrifty-prod, uses the Thrifty architecture and was trained on productive sequences and evaluated under the NT framework.

**NT-AA framework.** We also modeled affinity maturation more explicitly by combining a model to describe SHM and another model to describe selection. Specifically, the output from the selection model was applied as a "selection factor" to modify the predictions from the SHM model. The NT-AA framework offered a way to combine the ME perspective with the LM perspective to test if incorporating selection factors predicted by generic protein-based models could improve performance. For analyzing human repertoire data, we considered S5F and Thrifty-SHM each in combination with selection factors computed by ESM-1v, a general protein language model, or BLOSUM62, an amino acid substitution matrix. For the Replay project, we also combined ReplaySHM with a selection model based on deep mutational scanning (DMS) data that quantifies antigen binding for all single substitution mutants of the naive BCR.

**AA framework.** Lastly, models that only depend on amino acid sequence context were evaluated with the AA framework. These included language models AbLang2 and ESM-1v. AbLang2 is a MLM trained on productive BCR sequences.

We chose AbLang2 because it is the state-of-the-art model that used a focal loss to escape the trap of overfitting to the germline sequence, and thus we expected it to do better than other antibody language models for mutation prediction. However, by virtue of being an MLM, it predicts antibody evolution without disentangling the effects of SHM and selection. ESM-1v is a general protein MLM trained on a large database of proteins. Antibody sequences make up only a tiny fraction of the ESM-1v training data, so we interpreted the predictions of ESM-1v as amino acid preferences without any influence from SHM. Nevertheless, we also evaluated the performance of standalone ESM-1v to model affinity maturation because it has been used in the literature for antibody engineering [36].

## Site substitution probability was best described by nucleotide-based models

We computed the overlap and R-precision metrics to evaluate the ability of models to predict sites of amino acid substitution. For the two SHM models, Thrifty-SHM (Fig 2A, top) performed better than S5F (Fig A in S1 Text, top left) on both metrics (respectively, 0.834 versus 0.784 on overlap and 0.127 versus 0.0963 on R-precision). Comparing the observed and expected distributions for S5F, disagreements between observed and expected tended to be overestimation of the number of substitutions at sites in the framework regions (FWRs) and underestimation in the complementarity-determining regions (CDRs). Meanwhile, Thrifty-SHM predictions gave noticeably better agreement.

Surprisingly, adding ESM-1v to provide selection factors via the NT-AA framework resulted in markedly poorer performance than having just an SHM model. Notably for Thrifty-SHM + ESM-1v (Fig 2A, middle) and S5F + ESM-1v (Fig A in S1 Text, bottom left), the substitution counts at many sites were greatly underestimated. The overlap and R-precision metrics for the combined models remained almost the same or became worse compared to the performance of either SHM model alone. To further understand this lack of agreement, we looked at the SSP distributions for ESM-1v (Fig B in S1 Text, top right), which plot observed and expected number of substitutions in bins of SSP values. ESM-1v expected many more substitutions at sites with higher SSP (>0.1) than was observed, while it expected too few at lower SSP. The situation was not improved when ESM-1v provided selection factors in combination with either SHM model (Fig A in S1 Text, bottom row). Meanwhile, using the BLOSUM62 matrix to generate selection factors had little impact on performance relative to S5F or Thrifty-SHM alone (Table D and Fig A in S1 Text).

The observed and expected distributions for AbLang2 (Fig 2A, bottom) showed overestimation of substitution counts in the CDR3, while underestimation of the counts at many sites elsewhere. The AbLang2 model had overlap and R-precision comparable to S5F. AbLang2 was also assigning high SSPs more often than the data would suggest, although to a lesser degree than ESM-1v (Fig B in S1 Text, bottom right).

Overall, we found that nucleotide-based models provided a more accurate description of SSPs than language models trained on proteins, even one trained exclusively on antibodies (Fig 2B). The best performing model on overlap (0.923) and R-precision (0.139) was Thrifty-prod (Fig 3A). Computing overlap restricted to each FWR and CDR, the SHM-only models performed much better in CDRs than in FWRs, while the trend is reversed for AbLang2 (Fig C in S1 Text). We attributed this to SHM and nucleotide context being more important in CDRs, while purifying selection and amino acid preferences are more important in FWRs. Overlaps for Thrifty-prod were relatively consistent across regions. For regional R-precision scores, AbLang2 performed similar to or better than Thrifty-SHM or S5F respectively, but over full sequences it was worse than Thrifty-SHM and equal to S5F (Fig C in S1 Text). This highlights the value of nucleotide context to predicting sites of amino acid substitutions. Our findings held true across all human data sets we analyzed (Tables B–E in S1 Text).

## Models struggled to predict lack of substitution at highly conserved sites

Highly conserved sites provide a test of how well models describe sites under strong purifying selection. We considered seven sites that have previously been identified as being under strong purifying selection (Fig 3A): C23 and W41 around CDR1 [33]; C104 and W118 around CDR3 [33]; R43, D98, and Y102 that form a "charge cluster" in FWR3 [10].

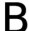

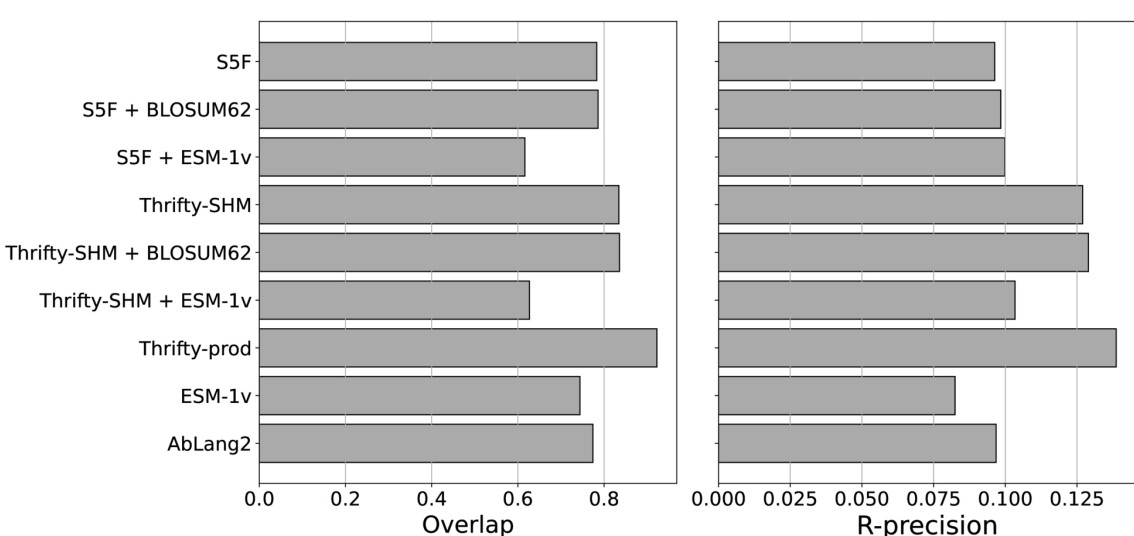

**Fig 2**. **Site substitution probability (SSP) was best described by models with nucleotide context.** (A) The observed counts distribution of amino acid substitutions per IMGT position over all PCPs in the data set (dots), and the expected counts distribution predicted by a model (bars). Overlap quantifies agreement between the observed and expected distributions. R-precision quantifies how often the observed sites of substitution in a PCP were also the most probable sites according to the model. The S5F model is an example in the NT framework, the Thrifty-SHM + ESM-1v model is an example in the NT-AA framework, and AbLang2 is an example in the AA framework. (B) The site substitution performance metrics on the Tang et al. data set for all models evaluuated.

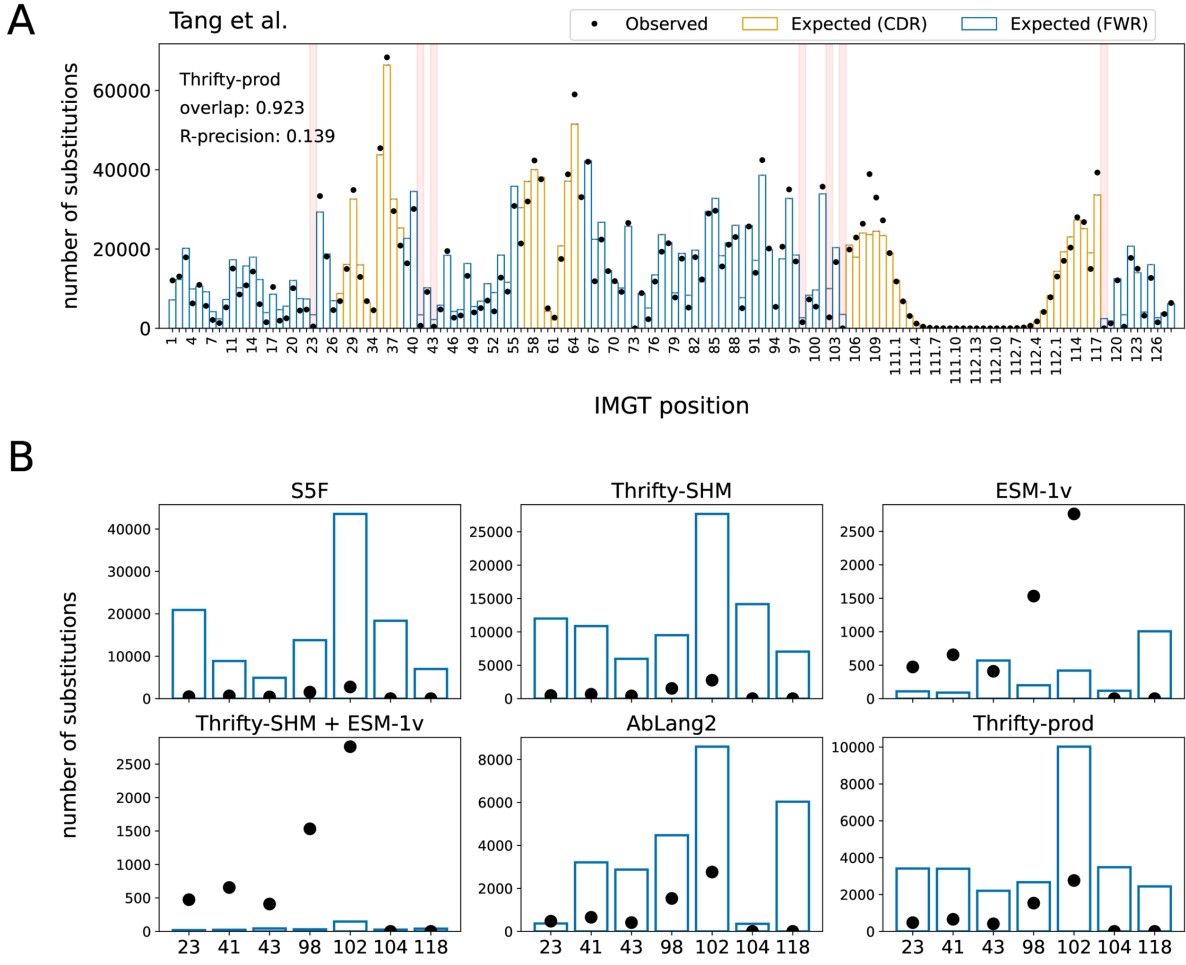

**Fig 3. Known highly conserved sites were not typically well described by models.** (A) The locations of seven highly conserved sites are highlighted in red. (B) Observed and expected number of substitutions at these seven conserved sites for various models.

We ranked sites by the observed and expected number of substitutions and focused on the relative rankings of these conserved sites in data versus model predictions. In the Tang et al. data set, the substitution counts observed at these conserved sites ranked relatively low among all positions; site Y102 was in the 25th percentile, site W41 was in the 19th percentile, and the other sites were below 15th. The S5F and Thrifty-SHM models, which were constructed to only describe SHM, substantially overestimated the number of substitutions at all seven sites (Fig 3B). These seven sites ranged in the 33rd to 98th percentiles in predicted number of substitutions for S5F, and 39th to 85th percentiles for Thrifty-SHM (Table G in S1 Text). The Thrifty-prod model also overestimated all sites, although to a lesser degree (22nd to 49th percentiles). AbLang2 provided relatively accurate estimates for sites C23 and C104, but overestimated the others. (We note that AbLang2 was trained exclusively on data with unmutated cysteines at these sites.) For these other sites, AbLang2 ranked them in the 25th to 46th percentiles. Finally, the Thrifty-SHM + ESM-1v model predicted these sites to be among the least substituted sites (9th to 10th percentiles), but other than sites C104 and W118, the model substantially underestimated the number of substitutions. It appeared that accurately inferring highly conserved sites in BCR sequences was a challenging task for models in both ME and LM perspectives.

PLOS Computational Biology

## Wide-context models trained on productive antibody sequences best predicted the identity of amino acid substitutions

We next evaluated the ability of each model to predict the correct amino acid at sites of substitution. Substitution accuracies (high is better) and CSP perplexity scores (low is better) were computed over all sites in the PCPs, as well as in each FWR or CDR region.

Overall, S5F and ESM-1v were the least accurate models. Better substitution accuracy was achieved when S5F was combined with a selection factor from BLOSUM62 or, interestingly, from ESM-1v (Fig 4A, 'All'). All other models had relatively similar accuracies, between 0.33 to 0.36, with Thrifty-prod being the most accurate. Both S5F and Thrifty-SHM were trained on data that did not involve selection effects. That Thrifty-SHM performed much better suggested that having a wider context—13-mer for Thrifty-SHM versus 5-mer for S5F—was helpful to predict amino acid outcomes.

Models involving ESM-1v had the worst CSP perplexity scores because their probability predictions were poorly calibrated with respect to the data (Fig 4B, 'All'). This applied to S5F + ESM-1v and Thrifty-SHM + ESM-1v as well, even though both models had substitution accuracies well above ESM-1v. Models involving Thrifty (with the exception of

**A**

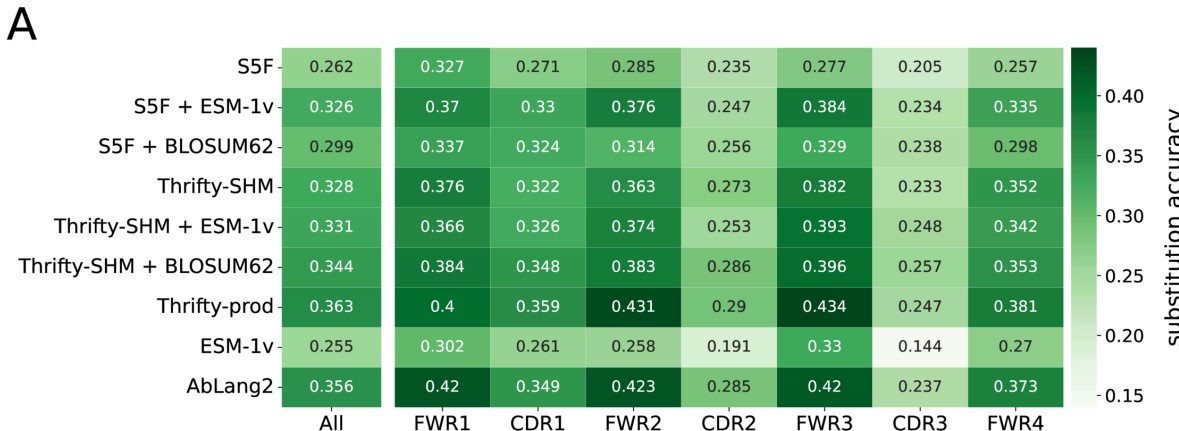

**B**

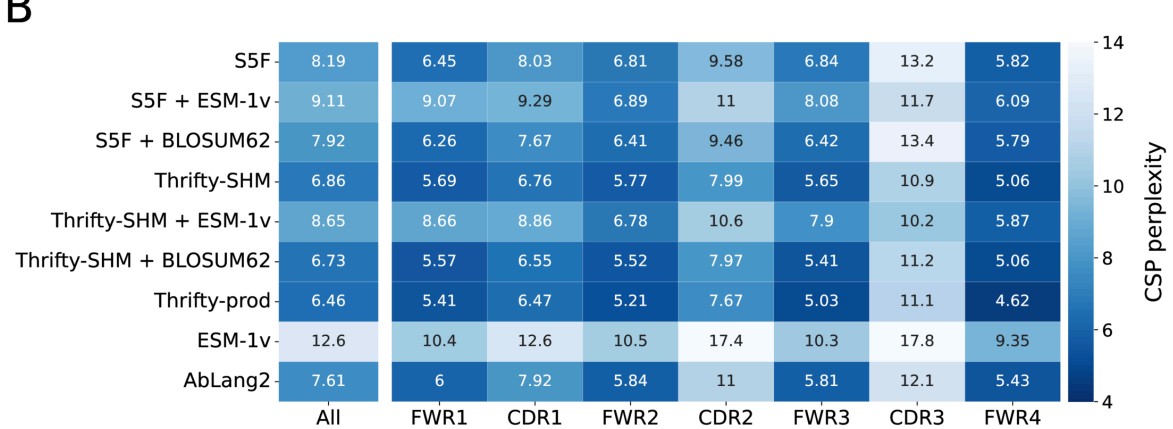

**Fig 4. Model performance metrics on amino acid predictions for substitutions.** Metrics include: (A) substitution accuracy, where high is better, and (B) CSP perplexity, where low is better. The left most column reports the evaluation over all sites in the sequences, and subsequent columns report the accuracy for sites in a specific region. A darker color indicates better performance.

Thrifty-SHM + ESM-1v) had the best CSP perplexity scores, better than AbLang2 despite not necessarily having higher substitution accuracies. The overall best CSP perplexity score came from Thrifty-prod.

We computed region-specific evaluations by restricting to sites in a FWR or a CDR. The general trend was that better relative performances were achieved in the FWRs than the CDRs (Fig 4). We interpreted this as being because sites in the FWRs tend to be more constrained by purifying selection and therefore easier to predict the outcome. It should be noted that there are many more sites in the FWRs than CDRs, so that FWR sites contributed more than CDR sites to the overall performance. These results were similar across all human data sets considered (Tables B–E in S1 Text).

We found that the relative ranking of models was broadly consistent when stratifying by number of substitutions per PCP (Figs D–G in S1 Text).

### Antigen-specific selection information offered modest improvements in performance over an SHM-only model

We now shift away from examining overall affinity maturation across diverse individuals and infection histories, and towards a specific setup with a known antigen. The Replay project [32] studied a mouse model of affinity maturation where all responses were initiated from the same naive BCR sequence exposed to the same antigen. Individual germinal centers were extracted and sequenced using single cell isolation.

The Replay project also provided ingredients to form a tailored model of affinity maturation. The ReplaySHM model was derived from analysis of non-productive "passenger" BCR sequences and predicted per-site SHM probabilities under the naive sequence context. In addition, a deep mutational scan was performed on all antibody variants with a single amino acid substitution away from the naive, yielding antigen-binding predictions that we used to compute selection factors. Neither of these ingredients used the germinal center data itself. We combined these together to form an explicit SHM-selection model, "ReplaySHM + DMS". These data are available for both IgH and Ig$\kappa$ (but not Ig$\lambda$), allowing for both heavy-chain- and light-chain-specific models to be implemented and evaluated. Our expectation was that this model, derived from bespoke measurements, could achieve close to the best possible performance.

We evaluated on the Replay data using ReplaySHM alone, combining ReplaySHM with DMS, BLOSUM62, or ESM-1v, and using AbLang2 (Figs 5 and 6). The observed and expected substitution counts distributions are provided in the supplementary (Figs H and I in S1 Text). ReplaySHM + DMS did in fact give the best performance for predicting the location of substitution (Figs 5A and 6A). For substitution accuracy (Figs 5B and 6B) and CSP perplexity (Figs 5C and 6C), ReplaySHM + DMS and ReplaySHM + BLOSUM62 performed similarly as the best models. Interestingly, AbLang2 achieved by far the highest substitution accuracy in the CDR3 in both IgH and Ig$\kappa$ data, but did relatively poorly in all other regions. It also had relatively poor CSP perplexity scores in most regions.

The addition of DMS selection factors improved upon ReplaySHM, demonstrating that our NT-AA framework produced an effective combination when suitable models for SHM and selection were integrated. On the other hand, ReplaySHM alone performed better than AbLang2, recapitulating the finding with human repertoire data that accurate SHM modeling was more effective at predicting affinity maturation than a state-of-the-art antibody language model.

For the Replay data set, the ReplaySHM + DMS model performed the best, as expected. However, the gains from incorporating binding measurements to ReplaySHM were modest. We attempted to incorporate expression measurements to calculate selection factors and found the same results as with binding alone (Table H in S1 Text). Splitting the data into PCPs where the naive BCR sequence is the parent and PCPs where the child sequence is a leaf node provided some insight (Table I in S1 Text). Performance gains for ReplaySHM + DMS over ReplaySHM alone is substantially greater in the naive-parent PCPs than in the leaf-child PCPs for substitution accuracy and CSP perplexity in both heavy and light chain data, and for R-precision in light chain data. Note that less than 10% of all PCPs are naive-parent PCPs but over 70% are leaf-child PCPs. The DMS measurements were relative to the naive BCR sequence and therefore should be more accurate when applied to naive-parent PCPs than for leaf-child PCPs. Furthermore, leaf node B cells may not have been fully subjected to the selection process [32], consequently selection factors would play a lesser role than

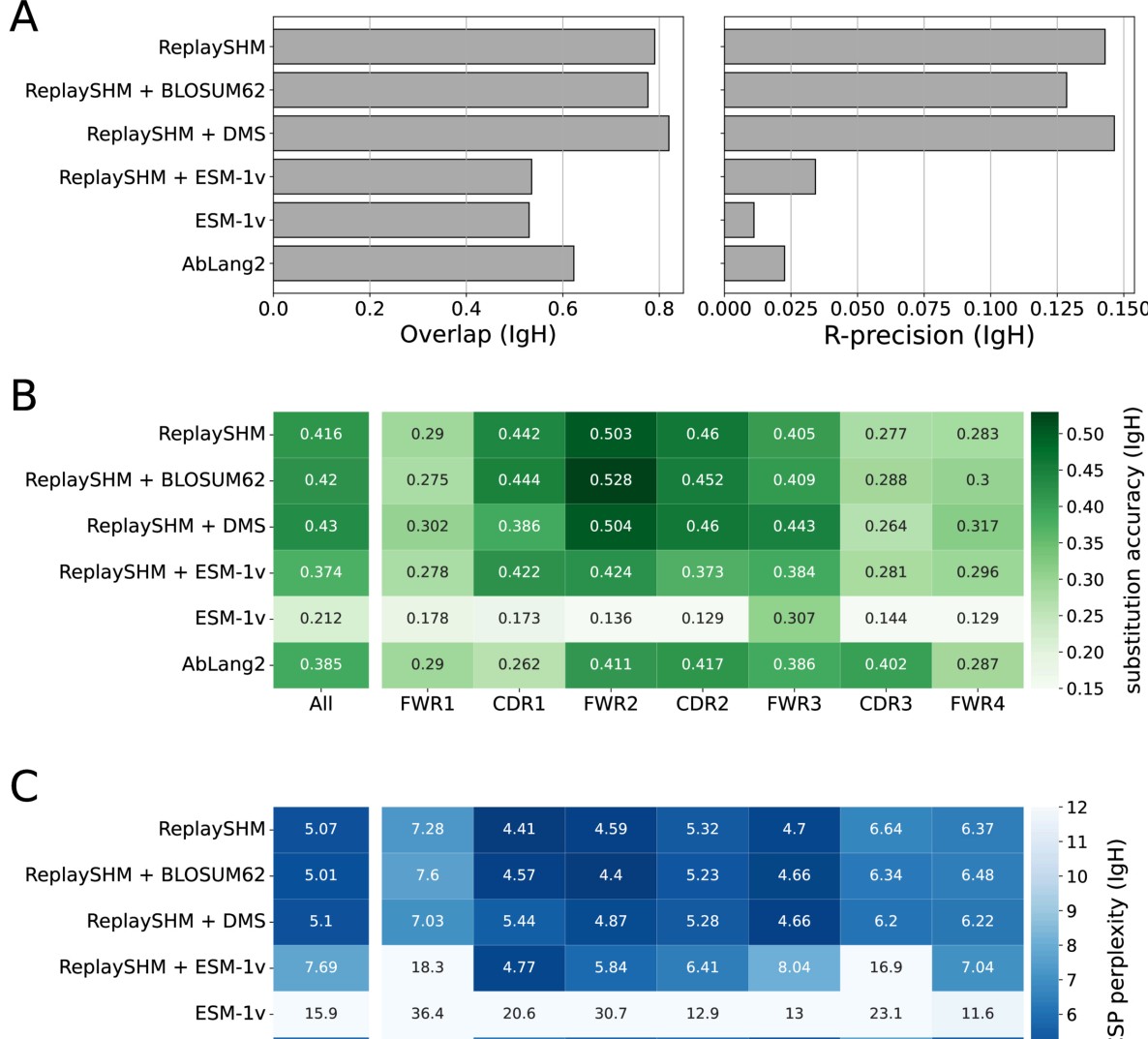

**Fig 5. Performance of models on Replay heavy chain data.** (A) Site substitution overlap and R-precision. (B) Substitution accuracy of the most probable amino acid at sites of substitution. (C) CSP perplexity.

SHM modeling for leaf-child PCPs. Indeed, we observe that ReplaySHM performs much better on leaf-child PCPs than on naive-parent PCPs in almost all cases (the exception being R-precision for heavy chain data). Interestingly, the overlap and R-precision for heavy chain data show no gains for naive-parent PCPs and only small gains for leaf-child PCPs, suggesting the possibility that ReplaySHM already captures much of the predictive power for which sites will undergo substitution.

We expected that having a model tailored for data from a highly controlled experiment could provide an estimate of upper-bounds on our performance metrics in general. For substitution accuracy and CSP perplexity, ReplaySHM + DMS performed better on both metrics in heavy chain (Table J in S1 Text) and light chain data (Table K in S1 Text) compared with the best of any model on human repertoire data we evaluated (Tables B–E in S1 Text). For R-precision, the Replay

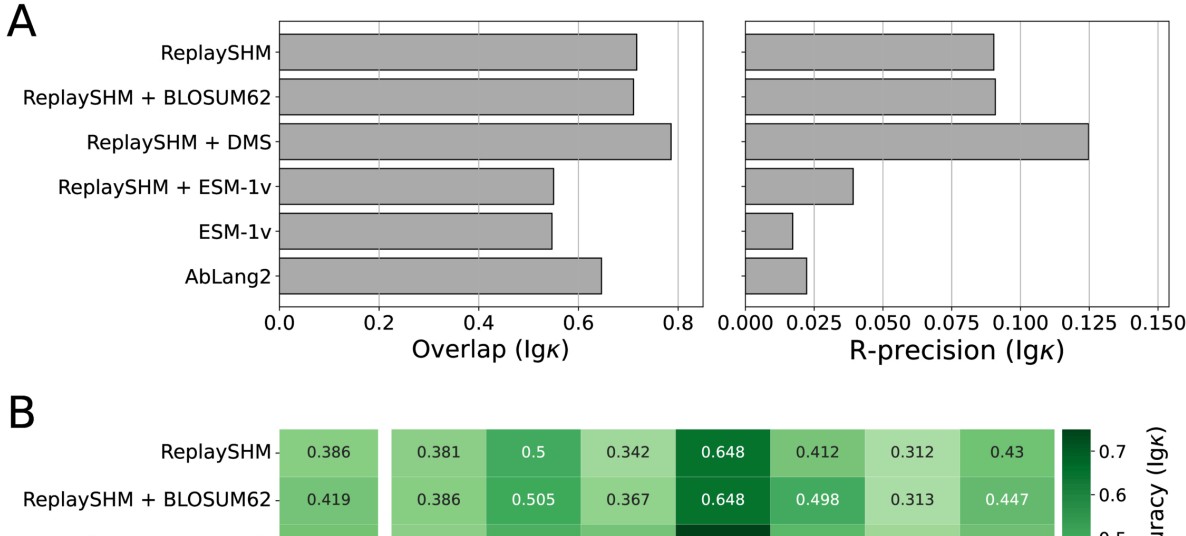

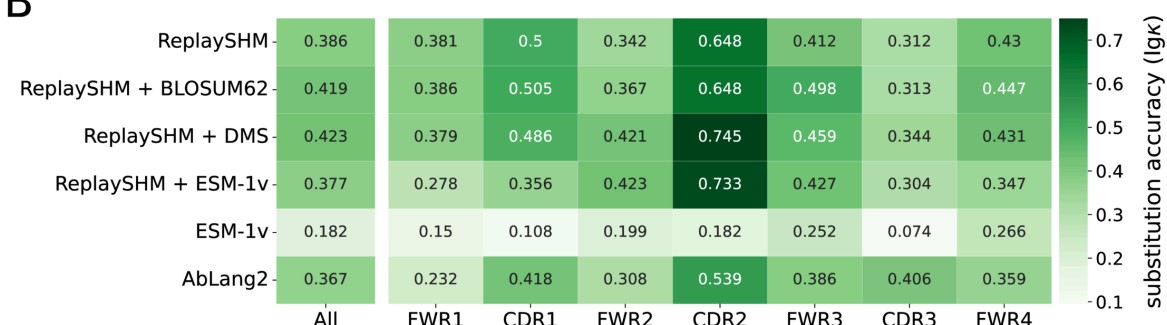

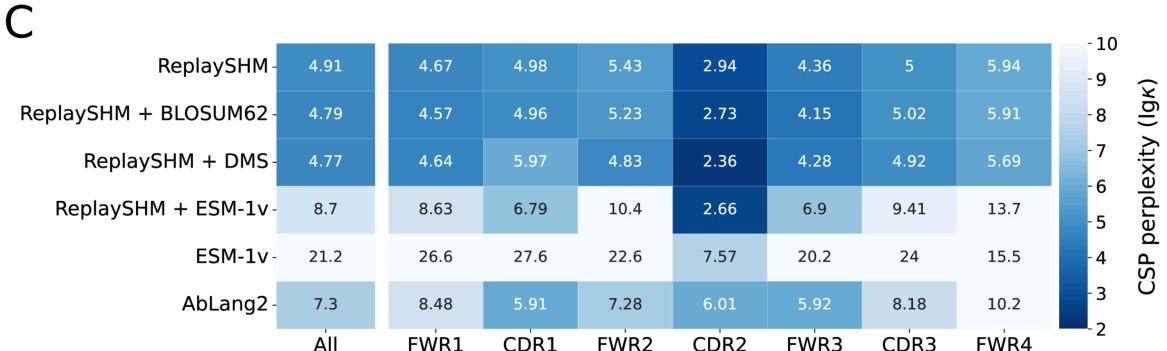

**Fig 6. Performance of models on Replay light chain data.** (A) Site substitution overlap and R-precision. (B) Substitution accuracy of the most probable amino acid at sites of substitution. (C) CSP perplexity.

performance was also better, provided that the data was stratified by number of substitutions (Fig J in S1 Text). This is because R-precision varies strongly on the number of substitutions per PCP, and the human repertoire data had both higher medians and maximums than the Replay data. On the other hand, the best overlap achieved with our Replay models was lower than those from Thrifty-prod.

## Discussion

Researchers have used a diversity of approaches to model antibody sequences. The ME models of affinity maturation directly probe the factors governing antibody evolution via estimating a limited set of parameters specific to the biological

process. These models are usually evaluated on how the inclusion of parameters improves the likelihood of the reconstructed evolutionary process [17–19]. On the other hand, MLMs use a large number of parameters trained to model the sequences themselves, and are evaluated based on their pseudo-log-likelihoods and perplexity on sequence data. They have also demonstrated some ability to predict functional properties useful for antibody engineering [25,28,36–38]. The MLMs do offer indirect insights into affinity maturation because antibody function reflects the outcome of its evolution. Thus although there is overlap in the goals of the ME and LM perspectives, little effort has been made to compare or combine the ME and LM techniques.

We developed EPAM to have a unified framework in which we could evaluate the performance of the complete spectrum of models to predict BCR affinity maturation. We analyzed the branches of phylogenetic trees describing B cell lineages and generated model predictions that involve branch length optimization to account for evolutionary time between clones. We chose to evaluate models using metrics common for classification tasks, rather than using metrics such as likelihood under a continuous-time Markov chain (CTMC) model of sequence evolution. Our justification in doing so is simply that not all the models we evaluated fit into the CTMC framework, however every CTMC model can be evaluated using the metrics used here. Our approach to calculate probabilities of amino acid outcomes from nucleotide-based predictions of SHM has similarities to the ARMADiLLO framework [39]. That approach generates predictions under the S5F model of SHM while matching the number of mutations in a PCP, while we match branch lengths. Because branch lengths take rate into account, this should result in slightly better performance for the NT framework. Prior efforts have been made to benchmark language models [22,37], but we exclusively focus on the ability of these models to thoroughly describe affinity maturation. Others have also evaluated antibody protein language models on predictions along B cell lineages [38], but do not account for evolutionary time. Furthermore, these works have focused exclusively on the LM perspective.

In human repertoire settings and in an antigen-specific mouse experiment we found that SHM-only models (Thrifty-SHM and ReplaySHM) outperformed MLMs (AbLang2 and ESM-1v) on almost all metrics. Our analysis quantified the dominant influence of the "SHM machinery" on BCR evolution, extending the results of Sheng et al. [11]. In contrast, MLMs lack an explicit consideration of nucleotide sequences and SHM, although recent language modeling efforts have sought to incorporate effects from affinity maturation more explicitly. Focal loss was introduced to the training of AbLang2 [28], leading to modest improvements over AbLang1 on our performance metrics (Fig K in S1 Text). Another approach incorporated a mutation position prediction (MPP) objective function into fine-tuning [22]. In spirit, we expect that the MPP function is similar to our R-precision and substitution accuracy metrics and may capture similar information regarding SHM and antibody evolution. Unfortunately, we were unable to include the EATLM model (or others fine-tuned with the MPP function) in our analysis due to the lack of publicly available trained model weights [22].

We also evaluated models that can be expressed as a combination of SHM and selection components. With an appropriate model for fitness, such as the DMS binding measurements for the Replay experiment, the combined model brought improvement over the SHM-only model. However, incorporating ESM-1v to generate selection factors had substantially reduced performance on most metrics, suggesting ESM-1v is not a good model of amino acid preferences for antibodies. It may be that using the ratio of ESM-1v amino acid probabilities to determine selection factors (see Methods) is not the right choice. Indeed, in the context of the Replay experiment, we see weak correlation between the ESM-1v selection factors and the DMS measurements (Fig L and Table L in S1 Text). Generally, ESM has been shown to be only moderately predictive of DMS measurements for a wide array of proteins [40] and often does not predict high fitness DMS variants [36].

In our evaluations on human repertoire data sets, we found Thrifty-prod to be the best model overall. The Thrifty-prod model is something of a hybrid—it does not explicitly involve amino acid preferences, but its performance suggests a better description of affinity maturation can be attained by building on an SHM-only model with proper accounting for selection effects. Drawing from our NT-AA framework formulation, one approach could be to train a protein sequence language

model to learn selection factors from the "residual" effects not described by SHM predictions. This way, we can leverage the strengths of LM techniques and involve the nucleotide context important for SHM modeling.

Overall, although ReplaySHM + DMS is certainly a good model, we were surprised that it did not outperform human models on repertoire data by a more significant margin. It is possible that this comes from the fact that our human models were evaluated using a diversity of naive sequences, which led to an averaging effect that improved overall measures of accuracy. One shortcoming of the ReplaySHM and DMS measurements was that they were relative to the naive BCR sequence, and becomes less accurate when applied over the course of affinity maturation.

None of the models we evaluated explicitly incorporate structural information, although masked language models have demonstrated some ability to learn structural constraints from sequence alone [41,42]. Our empirical analysis at highly conserved residues reveals that this implicit learning is insufficient: while AbLang2 and ESM-1v perform better than SHM-only models, they still substantially overestimate substitutions at structurally constrained sites such as W41, W52, and W118. Unsurprisingly, combining these models with mutation models does not successfully capture structural constraints either, performing worse than protein language models at conserved sites.

We hope the performance assessments tabulated in this work offer useful benchmarks to evaluate future antibody models to describe affinity maturation or other antibody properties.

## Materials and methods

The two major analysis steps were data preparation (Fig 1A) and model performance evaluation (Fig 1B and 1C). Data preparation involved separating sequences of a BCR repertoire into clonal families, where sequences likely to be descendants from the same naive B cell were clustered together. This was followed by inferring phylogenetic trees and ancestral sequences to reconstruct the course of affinity maturation along these clonal lineages. The branches of the trees were the inputs for performance evaluation, describing affinity maturation events where the "parent" sequence evolved to the "child" sequence. Amino acid substitution probabilities for a model were computed based on the parent sequence and the branch length that maximized the model's likelihood for the child. The predictions were compared against the substitutions that appeared in the child sequence.

### Data

**Data preparation.** Our pipeline began with pre-processed, full-length BCR DNA sequences, obtained from public repositories or directly from authors. The data sets used here were pre-processed by the original authors using pipelines specific to the respective sequencing technologies. Generally, the pre-processing steps include the following: assembling paired-end reads, filtering out low-quality sequences, and generating consensus sequences for studies with UMIs. These data sets each include BCR sequences from the repertoires of multiple individuals and are described in later sections. Here, we describe the procedure of processing a data set of observed sequences into a set of parent-child pairs (PCPs) for model evaluation following the strategy of Spisak et al. [14] (Fig 1A).

First, we performed germline inference and clonal family clustering using `partis` [43–46]. Germline gene annotations are based on OGRDB human germline sets IGH_VDJ ver 7 [47], IGKappa_VJ ver 2 [48], and IGLambda_VJ ver 1 [49]. Clustering on paired heavy and light chains was done if paired information is available. For each clonal family, we kept only productive sequences that did not have inferred insertions/deletions (indels). Explicitly, in `partis` terms: the `in_frames` flag was True, the `stops` flag was False (no stop codons), the `mutated_invariants` flag was False (the highly conserved codons that bookend the CDR3 must not differ from the inferred germline), and the `indel_reversed_seqs` was empty (empty because it did not differ from the observed input sequence). To further reduce spurious mutations from sequencing errors, we removed sequences that had more than 10 nucleotide mutations relative to the inferred naive sequence in any window of 20 consecutive sites. While this may remove sequences that are truly highly diversified, for instance in the CDRs, it should have little impact on our analysis because the overlap, substitution accuracy, and CSP

perplexity metrics showed mild variation at high number of substitutions (Figs D, F, and G in S1 Text). We then performed phylogenetic inference and ancestral sequence reconstruction for each clonal family. We ran `IQ-TREE` [50] with a K80 substitution model using the naive sequence as outgroup—following the approach of Spisak et al. [14], and the FreeR-ate site heterogeneity model. For data sets with paired heavy chain and light chain information, we ran the *edge-linked-proportional* partition model (`-p` option) that gives each chain its own evolutionary rate [51], motivated by the analysis on phylogenies inferred with paired information reported in Jensen et al. [52]. The edges of the inferred phylogenetic trees were taken as PCPs. The first site of the sequence was set to align with the start of the V gene; if necessary, nucleotides before the start of the V gene were truncated or sites at the 5' end with missing reads were padded with `n`. Because we were interested in amino acid substitutions, the sequence length was restricted to be a multiple of 3 by truncating sites at the 3' end, if necessary. We dropped any PCP that contained a stop codon; these can arise from the inference of ancestral states.

For model evaluation, we restricted to clonal families where all BCR sequences were full-length and without ambiguous nucleotides. We considered PCPs from branches up to the most recent common ancestor (MRCA). We filtered out PCPs with nucleotide substitution frequencies greater than 30% relative to their sequence length, as these sequences were highly diverged and may have indicated an issue with a particular branch. Each sequence in a parent-child pair was translated to its amino acid sequence using `Biopython` [53]. We only considered PCPs with at least one amino acid substitution between parent and child sequences. Table A in S1 Text summarizes the clonal family sizes and PCP counts for all data sets.

For aligning sequences in human repertoire data sets, we derived antibody numbering using ANARCI, restricting our search space to human species and the appropriate antibody chain [54]. We selected a representative sequence from every clonal family and ran ANARCI to assign IMGT numbering to all representative sequences in a data set. This representative sequence was generally the naive sequence for the clonal family. In some cases, the inferred naive sequence may contain a stop codon, in which case we selected the next most ancestral sequence. We then used the IMGT numbering to align all sequence for visualization and further analysis. We followed the IMGT conventions for defining CDR and FWK regions (see Table 1). Very rarely, ANARCI returned numberings that infer unusual insertions (represented by decimal numbers outside of sites 111–112 in CDR3), or were inconsistent with the observed sequence length. In both of these cases, the sequences in that clonal family were dropped from analysis.

**Tang et al. data.**  Non-productive human IgH sequences from the Tang et al. data set were previously used for modeling SHM [15]. Originally generated by [55,56], this data set included full-length BCR repertoires from 21 individuals. These BCR sequences were pre-processed using the `pRESTO` toolkit for assembly, quality control, and UMI-error correction [57]. We obtained the pre-processed sequences directly from the authors. From each individual sampled, we extracted productive IgH DNA sequences from marginal zone (MZ), memory (M), and plasma (PC) B cells, and processed them using our `partis-IQ-TREE` pipeline described above. We note that at least some of the sequences in this data set (the samples originally generated in [56]) are included in the Observed Antibody Space (OAS) database used to train AbLang models [3,5,21,28].

**Jaffe et al. data.**  The Jaffe et al. data set consisted of productive, human, paired IgH and Ig$\kappa$/Ig$\lambda$ sequences generated by 10x Genomics [31]. This data set included full-length BCR repertoires from four individuals. We downloaded the publicly available pre-processed sequences from Figshare and filtered for memory cells. This data set was used to train Thrifty-prod, and is also included in the OAS database.

**Table 1**. **Definitions of FWR and CDR regions.** IMGT positions for each FWR and CDR region.

| FWR1 | CDR1 | FWR2 | CDR2 | FWR3 | CDR3 | FWR4 |
|------|------|------|------|------|------|------|
| 1–26 | 27–38 | 39–55 | 56–65 | 66–104 | 105–117 | 118–128 |

**Rodriguez et al. data.** The Rodriguez et al. data are productive human IgH sequences obtained with the 5' RACE-Seq method [30]. This data set included full-length BCR repertoires from 51 individuals. We received the sequences directly from the authors following `pRESTO` processing (assembly, quality control, and UMI-error correction). This data is not currently included in OAS database, and thus can serve as a true out-of-sample evaluation for such models.

**Ford et al. data.** The Ford et al. data are productive human IgH sequences obtained using the FLAIRR-Seq protocol [29]. This data set included full-length BCR repertoires from 10 individuals. We received the sequences directly from the authors following `pRESTO` processing (assembly, quality control, and UMI-error correction). While this data set was relatively small, it was one of the few data sets that includes full-length BCR sequences from healthy individuals, and thus was a valuable resource for evaluating models of affinity maturation. This data is not currently included in OAS database, and thus can additionally serve as a true out-of-sample evaluation for such models.

**Replay data.** The Replay experiment [32] explored affinity maturation in a mouse model with a specific naive BCR sequence under challenge from a specific antigen. Mice were engineered to be unable to form germinal centers with their own B cells. B cells that are specific to the known antigen were transferred to the mice, and germinal centers form when they were immunized with the antigen. These donor B cells underwent affinity maturation. Unlike with the other data sets, the phylogenetic inference was performed with the GCTree [58,59] method because these sequences were so close to naive. While this data set is much smaller than the human repertoires we analyzed, it came from a highly controlled experimental setting.

## Methods

To compare the performance of various models, we needed to compute a consistent set of quantities while handling the dissimilar ways different models are used. Given a model, the fundamental calculations were the per-site amino acid probabilities from the parent sequence of a PCP. These probabilities had to scale with the amount of evolutionary "time" separating parent to child. Therefore, in calculating the probabilities, we calibrated the branch length by optimizing the likelihood for the model to describe the PCP. The amount of evolutionary time became dependent on the model being evaluated and the predicted probabilities corresponded to the best fit to data. This value, $\hat{\tau}$, varied widely across models for a given PCP (Fig M in S1 Text). Optimized values for Thrifty-SHM and Thrifty-prod were closest to the branch length inferred by `IQ-TREE`. This design choice allowed all models to be evaluated in their best light.

Below, we describe the calculations to derive per-site amino acid probabilities and the models associated with the different frameworks. Our notation denotes nucleotide bases by lower case letters and amino acids by capital letters.

### Probability of substitution in the NT framework

This framework takes a model that generates output per nucleotide site, and groups them by codon to calculate amino acid probabilities. The nucleotide-based models we analyzed report a neutral mutation rate $\lambda_i$ and a vector of conditional mutation probabilities $(p_{i,\text{a}}, p_{i,\text{c}}, p_{i,\text{g}}, p_{i,\text{t}})$ for each nucleotide site $i$. The probabilities are conditioned on a mutation occurring, so the wildtype nucleotide probability is zero (note this is different than the typical assumption for continuous-time Markov chain evolutionary sequence models). We combined mutation rate and probabilities together to get a per-site model assuming exponential waiting time until mutation. Therefore, the probability for a site to remain unchanged after time $\tau$ is $e^{-\lambda_i \tau}$. The probability of a given nucleotide mutation, $P_{i,\text{x}}(\tau)$, from wildtype to non-wildtype $\text{x}$ is

$$P_{i,\text{x}}(\tau) := \left(1 - e^{-\lambda_i \tau}\right) p_{i,\text{x}}.$$

For a given PCP, the value of $\tau$ was determined by maximizing the likelihood of the child sequence

$$\mathcal{L}(\tau) = \prod_i P_{i,\text{x}_i}(\tau), \tag{1}$$

where $x_i$ were the nucleotides of the child sequence.

To obtain amino acid probabilities, we considered the codons encoded by the nucleotide sequence and summed over all possible mutations that led directly to a particular amino acid. For example, suppose codon $\mathbf{r}$, corresponding to nucleotide sites $(r_0, r_1, r_2)$, encoded `ggt` (glycine) in the parent sequence. The probability for cysteine (`tgt` and `tgc`) would be $\mathbb{P}_{\mathbf{r},\texttt{tgt}} + \mathbb{P}_{\mathbf{r},\texttt{tgc}}$, where

$$\mathbb{P}_{\mathbf{r},\texttt{tgt}}(\tau) = P_{r_0,\texttt{t}}(\tau)P_{r_1,\texttt{g}}(\tau)P_{r_2,\texttt{t}}(\tau) = \left(1 - e^{-\lambda_{r_0}\tau}\right)p_{r_0,\texttt{t}}\left(e^{-\lambda_{r_1}\tau}\right)\left(e^{-\lambda_{r_2}\tau}\right)$$

and

$$\mathbb{P}_{\mathbf{r},\texttt{tgc}}(\tau) = P_{r_0,\texttt{t}}(\tau)P_{r_1,\texttt{g}}(\tau)P_{r_2,\texttt{c}}(\tau) = \left(1 - e^{-\lambda_{r_0}\tau}\right)p_{r_0,\texttt{t}}\left(e^{-\lambda_{r_1}\tau}\right)\left(1 - e^{-\lambda_{r_2}\tau}\right)p_{r_2,\texttt{c}}.$$

The amino acid probability for the wildtype, in other words for no substitution, is the complement to the sum of probabilities from all non-wildtype outcomes,

$$1 - \sum_{x_0,x_1,x_2 \text{ not all WT}} \mathbb{P}_{\mathbf{r},x_0x_1x_2}(\tau).$$

Note that we are conditioning on a non-stop codon outcome, as sequences with stops were excluded from our analysis.

This approach enabled us to use context-sensitive mutation rates for the parent sequence. It is well established that mutation rates in SHM depend on nucleotide patterns surrounding a site, and recently developed tools on amino acid preferences for antibodies or proteins in general make predictions based on the full sequence context. However, this approach has limitations, such as that it cannot account for multiple mutations that may occur at a site as the parent evolves to the child. In the case of BCRs, the approximation is reasonable because branch lengths of PCPs are typically short, therefore multiple mutations should be very rare. We confirmed that the expected number of mutations for PCPs under NT models was far below one mutation per site for all human data sets considered (Fig N in S1 Text). In theory, an alternative would be to approximate a sequence-valued continuous-time Markov chain (e.g. [60]) but the Thrifty models use a very wide context and adding selection as described below is truly a sequence-valued process. The models that follow the above formulation are Thrifty, S5F, and ReplaySHM.

**Thrifty models.**  Thrifty is a CNN-based approach designed to describe SHM [16]. The Thrifty-SHM model was trained on PCPs from non-productive BCR sequence data that were processed through clonal family clustering and phylogenetic inference. The same Thrifty architecture was trained on productive sequences so that the mutation rates and probabilities implicitly involved SHM and selection effects. The Thrifty-prod model is the Thrifty architecture trained on the Jaffe et al. productive data. Branch lengths were estimated by optimizing Eq 1, with initial values of $\tau$ set to the normalized nucleotide mutation frequency (number of mutations in the PCP divided by the sequence length).

**S5F.**  The human heavy chain S5F model is a model for SHM derived from counting occurrences of mutation at four-fold synonymous sites in productive BCR sequences [13]. It makes predictions for the central site in 5-mer nucleotide motifs. Branch lengths were estimated by optimizing Eq 1.

**ReplaySHM.**  As part of the Replay project, non-productive BCR sequences were engineered by introducing indels to the heavy chain or light chain allele corresponding to the donor B cells. These sequences evolved due to SHM only as "passengers" in B cells that underwent affinity maturation. Mutation rates and conditional probabilities were measured with respect to the naive wildtype BCR sequence. For each chain subunit, the per-site mutation rates were parametrized using softmax and therefore the rates over all sites summed to 1. Branch lengths were estimated by optimizing Eq 1.

## Probability of substitution in the NT-AA framework

To combine nucleotide mutation rates and amino acid preferences, we introduced a selection factor, $F_{\mathbf{r},\mathcal{X}}$ (of an amino acid $\mathcal{X}$ at site $\mathbf{r}$), and modified the formulation from the preceding section,

$$\mathbb{P}_{\mathbf{r},x_0 x_1 x_2}(\tau) = P_{r_0,x_0}(\tau) P_{r_1,x_1}(\tau) P_{r_2,x_2}(\tau) F_{\mathbf{r},\mathcal{X}(x_0 x_1 x_2)},$$

where $\mathcal{X}(x_0 x_1 x_2)$ is the amino acid of the codon outcome. As before, the probability of no substitution is set to 1 minus the sum of the probabilities of all the non-wildtype outcomes, so that these $\mathbb{P}_{\mathbf{r},x_0 x_1 x_2}(\tau)$ form a probability distribution on codons. Thus we used the following likelihood to obtain $\tau$:

$$\mathcal{L}(\tau) = \prod_{\mathbf{r}} \mathbb{P}_{\mathbf{r},x_0 x_1 x_2}(\tau), \tag{2}$$

where $x_0 x_1 x_2$ now denotes the codon at site $r$ of the child sequence.

The selection factors generally depend on the parent amino acid sequence context. We quantified amino acid preferences from outputs of the ESM-1v model, entries from the BLOSUM62 matrix, or DMS measurements for Replay, by making the ratio, $R$, of non-wildtype to wildtype model outputs. These ratios can sometimes be very large, and cannot be used directly as selection factors because the resulting calculation for $\mathbb{P}_{\mathbf{r},x_0 x_1 x_2}$ might become larger than one. To mitigate this, we applied a transformation to $R$ as follows,

$$F = \frac{2}{1 + \frac{1}{R}}. \tag{3}$$

Therefore, when $R$ is close to one, the selection factor is also close to one. When $R$ is very large, the selection factor approaches two. In the rare cases where the resulting $\mathbb{P}_{\mathbf{r},x_0 x_1 x_2}$ was greater than one, we clamped the value at one. Whenever the sum of codon probabilities was larger than one, we normalized over all non-wildtype codon outcomes and assigned the probability of zero for the codon to remain unchanged. Model-specific details are described in the following relevant sections.

**Using DMS selection factors in the NT-AA framework.** A deep mutational scan was performed to obtain measurements of the binding strength, $-\log_{10} K_D$, between the antigen with every possible single substitution mutant to the naive Replay BCR. Therefore, these were estimates of the change in binding strength associated with each amino acid change. The dissociation constant ratio, $R = K_D^{\text{parent}}/K_D^{\text{child}}$, of parent to child amino acids was taken as $R$ in Eq 3. These selection factors were combined with the ReplaySHM to form an explicit SHM-selection model for the Replay data. Branch lengths were estimated by optimizing Eq 2.

**Using BLOSUM62-derived selection factors in the NT-AA framework.** We used the amino acid preferences described by the BLOSUM62 matrix [61] as another way to obtain selection factors. Briefly, the entries of the matrix are log odds ratios between two amino acids; positive values indicate a preference for the potential child amino acid over the parent, and negative values for the opposite. For parent and child amino acids, $\mathcal{X}'$ and $\mathcal{X}$ respectively, we took the corresponding BLOSUM62 matrix entry, $M(\mathcal{X}',\mathcal{X})$, and exponentiated to get the odds ratios, $R = 2^{M(\mathcal{X}',\mathcal{X})/2}$, to be used in Eq 3. (The factor of 2 is to accommodate the fixed scaling factor on log-odds ratios that's built into BLOSUM.) These selection factors are used in combination with S5F, Thrifty-SHM, and ReplaySHM. Branch lengths were estimated by optimizing Eq 2.

## Probability of substitution in the AA framework

We made use of several masked language models (MLMs) that have been trained to predict the likelihood of amino acid tokens at masked sites. Specifically, we evaluated antibody-specific models trained on OAS (AbLang1 and AbLang2) and

general protein models trained on UniRef90 (ESM-1v) [21,28,35,62]. These models were implemented at the protein level, which required us to translate parent sequences to amino acid sequences prior to making predictions. All the MLMs considered here have alphabets that are larger than the standard 20 amino acids, including tokens that account for ambiguous amino acids, gaps, or beginning/end of sequences. For our purposes, we were interested only in the probability of substitution from one unambiguous amino acid into another in a BCR sequence. We obtained the log likelihoods (or logits) directly from the MLMs for all tokens at each site, then restricted to the relevant tokens (corresponding to the 20 standard amino acids, $\mathcal{X}$) accordingly. Amino acid probabilities $P_{\mathbf{r},\mathcal{X}}$ were then obtained by:

$$P_{\mathbf{r},\mathcal{X}} = \text{softmax}(\mathcal{L}_{\mathbf{r},\mathcal{X}}),$$

where $\mathcal{L}_{\mathbf{r},\mathcal{X}}$ are the logits at site $\mathbf{r}$ for the 20 amino acids $\mathcal{X}$. The amino acid probabilities produced from MLM outputs are not directly comparable to those from SHM-based models as they are agnostic to evolutionary time $\tau$ and parent-independent in some cases. Our framework requires further transformation of these probabilities in order to perform a fair comparison on branches. The necessary steps are described here for the model types considered.

We supported two commonly used strategies to generate model likelihoods: 'masked-marginals' and 'wt-marginals'. All language model results presented in the main text used the masked-marginals strategy (as described below this is considered to be the more accurate version). A model with a wt-marginals strategy was provided the entire (unmasked) parent AA sequence and predictions were generated simultaneously using a single forward pass. These probabilities are parent-dependent, as they were computed relative to the wildtype (i.e. parent) amino acid at each site.

While the wt-marginals strategy is computationally efficient, performance improvements have consistently been observed for the masked-marginals strategy in evaluation tasks specific to both general MLM prediction and antibody engineering [35,36]. The masked-marginals strategy involved iteratively masking and predicting for each site in the parent sequence. This implementation required a forward pass through the model for each site in a given sequence, where a single site was masked at a time. In this case, the site-specific probabilities $P_{\mathbf{r},\mathcal{X}}$ are agnostic to the parent amino acid at that site (due to masking), and are thus parent-independent.

Still, the amino acid probabilities produced from MLM outputs thus far did not yet account for evolutionary time. We expected that these probabilities were likely to correspond to times much longer than those observed in our PCP branch lengths given their training schemes. We instead adjusted the MLM probabilities under a model in which there was an exponential waiting time until a substitution occurs. We do not have a direct way to convert these probabilities to rates, but we can scale the MLM probabilities by a factor $(1 - e^{-\tau})$, which is the mutability assuming Poisson probabilities. The amino acid probabilities under this model are then

$$\mathbb{P}_{\mathbf{r},\mathcal{X}}(\tau) = \begin{cases} e^{-\tau} + (1 - e^{-\tau})P_{\mathbf{r},\mathcal{X}}, & \text{if } \mathcal{X} \text{ is wildtype} \\ (1 - e^{-\tau})P_{\mathbf{r},\mathcal{X}}, & \text{if } \mathcal{X} \text{ is not wildtype} \end{cases}$$

The value of $\tau$ was determined by maximizing the likelihood of the child sequence under the model, where

$$\mathcal{L}(\tau) = \prod_{\mathbf{r}} \mathbb{P}_{\mathbf{r},\mathcal{X}_{\mathbf{r}}}(\tau) \tag{4}$$

and $\mathcal{X}_{\mathbf{r}}$ are the amino acids of the child sequence. We started with a value of $\tau$ that was the normalized nucleotide mutation frequency and optimized $\tau$ for each parent-child pair. This scaling approach allowed us to more properly compare AbLang and ESM-1v with SHM models that naturally incorporate branch lengths. The scaled amino acid probabilities did improve the agreement with data, but it did not change the relative probabilities so metrics like R-precision and

substitution accuracy were unchanged. Further details on specific models and their implementations are provided in the following sections.

**Using AbLang in the AA framework.** AbLang was trained with a masked modeling objective on the OAS database, which includes antibody protein sequences from multiple species and utilizes inferred germline sequences paired with observed antibody sequences. As is common for deep models, this training does not distinguish between SHM and selection processes nor does it include any notion of evolutionary time between germline and observed sequences. The training of AbLang2 additionally included a focal loss function designed to avoid germline bias found in earlier versions of the model [28]. We used both AbLang1 and AbLang2 as implicit SHM-selection models, as they have captured the joint effects of both processes on antibody sequences. Heavy and light chain sequences were considered independently in our framework and are fed into the AbLang models individually. For AbLang2, we considered both the masked-marginals and wt-marginals strategies for computing site-specific amino acid likelihoods $\mathbb{P}_{\mathbf{r},x}(\tau)$, and support heavy or light chains. Our AbLang1 model followed the wt-marginals strategy to obtain site-specific amino acid probabilities.

**Using ESM-1v in the NT-AA framework and as a standalone model.** We primarily considered ESM-1v as a source for selection factors to combine with SHM models, under the reasoning that ESM likely was not sensitive to SHM effects in its training. General protein MLMs like ESM-1v have learned general evolutionary principles in long evolutionary scale but more recent evidence highlights their limited knowledge of affinity maturation [63]. Previously, this model (as a full ensemble) has been used in combination with other ESM models for antibody engineering and other related tasks [22,28, 36]. So we additionally included it as a standalone model for completeness and direct comparison with the AbLang models. ESM-1v models were used in the NT-AA framework to produce selection factors, $F$, and the AA framework for amino acid probabilities, $\mathbb{P}_{\mathbf{r},x}(\tau)$.

EPAM supports the wt-marginals and masked-marginals scoring strategies for the full ESM-1v ensemble (models 1–5). The masked-marginals strategy was used for the standalone ESM-1v model as well as for computing selection factors in the NT-AA framework unless otherwise noted. We tried using the full ensemble in the NT-AA framework, running all five models independently and averaging their selection factors, $F$. For the standalone ESM-1v ensemble model, we averaged the site-specific amino acid probabilities output by each model pre-scaling, $P_{\mathbf{r},x}$. However, we saw no significant improvement in performance in either case over using an individual model alone, so an individual model was used alone in all main results for computational efficiency (Figs L and O in S1 Text). Specifically, we utilized model 1 of the five model ensemble, which we expected captures general, site-specific selection pressures for a given protein sequence context.

## Supporting information

**S1 Text. Supplementary materials.** File containing all supplementary tables and figures.
(PDF)

## Acknowledgments

We thank the following for sharing processed human data with us: Corey Watson, Easton Ford, Melissa Smith, Oscar Rodriguez, and other members of the Watson lab for providing the Ford et al. and Rodriguez et al. data sets, and Thomas MacCarthy for the Tang et al. data set. We thank members of the Matsen group as well as the Victora and DeWitt labs and Oxford Protein Informatics Group for helpful discussions.

## Author contributions

**Conceptualization:** Mackenzie M. Johnson, Kevin Sung, Hugh K. Haddox, Gabriel D. Victora, Yun S. Song, Julia Fukuyama, Frederick A. Matsen IV.

**Data curation:** Mackenzie M. Johnson, Kevin Sung, Ashni A. Vora, Tatsuya Araki, Gabriel D. Victora.

**Formal analysis:** Mackenzie M. Johnson, Kevin Sung, Frederick A. Matsen IV.

**Funding acquisition:** Yun S. Song, Frederick A. Matsen IV.

**Investigation:** Mackenzie M. Johnson, Kevin Sung, Ashni A. Vora, Tatsuya Araki, Gabriel D. Victora, Yun S. Song, Julia Fukuyama, Frederick A. Matsen IV.

**Methodology:** Mackenzie M. Johnson, Kevin Sung, Gabriel D. Victora, Julia Fukuyama, Frederick A. Matsen IV.

**Project administration:** Gabriel D. Victora, Frederick A. Matsen IV.

**Resources:** Frederick A. Matsen IV.

**Software:** Mackenzie M. Johnson, Kevin Sung, Frederick A. Matsen IV.

**Supervision:** Frederick A. Matsen IV.

**Validation:** Mackenzie M. Johnson, Kevin Sung, Frederick A. Matsen IV.

**Visualization:** Mackenzie M. Johnson, Kevin Sung, Hugh K. Haddox, Frederick A. Matsen IV.

**Writing – original draft:** Mackenzie M. Johnson, Kevin Sung, Frederick A. Matsen IV.

**Writing – review & editing:** Mackenzie M. Johnson, Kevin Sung, Hugh K. Haddox, Ashni A. Vora, Tatsuya Araki, Gabriel D. Victora, Yun S. Song, Julia Fukuyama, Frederick A. Matsen IV.

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
