## [Decision Letter · Decision Letter 0]

28 Aug 2025

PCOMPBIOL-D-25-01510

Nucleotide context models outperform protein language models for predicting antibody affinity maturation

PLOS Computational Biology

Dear Dr. Matsen IV,

Thank you for submitting your manuscript to PLOS Computational Biology. After careful consideration, we feel that it has merit but does not fully meet PLOS Computational Biology's publication criteria as it currently stands. Therefore, we invite you to submit a revised version of the manuscript that addresses the points raised during the review process.

Please submit your revised manuscript within 60 days Oct 28 2025 11:59PM. If you will need more time than this to complete your revisions, please reply to this message or contact the journal office at ploscompbiol@plos.org. Please include the following items when submitting your revised manuscript:

We look forward to receiving your revised manuscript.

Kind regards,

Alexey Onufriev

Academic Editor

PLOS Computational Biology

James Faeder

Section Editor

PLOS Computational Biology

**Additional Editor Comments:**

One of the critical comments to address is novelty of the proposed model; including a detailed comparison with ARMADiLLO, as proposed by one of the reviewers, makes sense.

**Journal Requirements:**

At this stage, the following Authors/Authors require contributions: Mackenzie M. Johnson, Kevin Sung, Hugh K. Haddox, Yun S. Song, Julia Fukuyama, Ashni A. Vora, Tatsuya Araki, Gabriel D. Victora, and Frederick A. Matsen IV. Please ensure that the full contributions of each author are acknowledged in the "Add/Edit/Remove Authors" section of our submission form.

3) Thank you for stating "Upon publication, processed data files will be made available on Zenodo." We strongly recommend all authors decide on a data sharing plan before acceptance, as the process can be lengthy and hold up publication timelines. Please note that, though access restrictions are acceptable now, your entire data will need to be made freely accessible if your manuscript is accepted for publication. This policy applies to all data except where public deposition would breach compliance with the protocol approved by your research ethics board. 

4) Please amend your detailed Financial Disclosure statement. This is published with the article. It must therefore be completed in full sentences and contain the exact wording you wish to be published.

5) Thank you for stating that "G.D.V. is an advisor for and holds stock of the Vaccine Company. T.A. is currently an employee of Pfizer Inc." Please amend your 'Competing Interests' statement and declare all competing interests beginning with the statement "I have read the journal's policy and the authors of this manuscript have the following competing interests:"

**Reviewers' comments:**

Reviewer's Responses to Questions

**Comments to the Authors:**

**Please note that one review is uploaded as an attachment.**

Reviewer #1: In there manuscript ”Nucleotide context models outperform protein language models for predicting antibody affinity maturation” Johnson et al. develop and benchmark a model taking nucleotide sequence context into processes for modelling of affinity maturation. This approach has scientific relevance as hypermutation is strongly influenced by the nucleotide sequence context both in terms of promoting or preventing substitution. The authors conclude that such an approach performs better that other models. The authors might however also consider the following aspects:

In the past, the ARMADiLLO tool (e.g. 10.1093/nar/gkad398) uses a similar concept to assess evolution of antibodies. How do these methodologies compare in terms of accuracy of evolution prediction?

Evolution does, as the authors predict, not only depend on the linear protein sequence but also on the nucleotide sequence that encodes it. However, evolution also depends on the structural context that neither methodology really approach. Some residues (e.g. but not limited to C41, W41, W52, and C104) are so important for the structure that they never evolve. Others may evolve but are restricted to evolution to conserved substitutions (e.g. L, I, M; nucleotide sequence prediction softwares (e.g. ARMADiLLO) would typically overestimate substitutions at such sites) while others are able to accommodate essentially any residue. Other parts, not specifically discussed, of the domains, such as the upper and lower cores, may have similar restrictions. In some cases, germline genes encode unusual resides (not encoded by other genes, such as residues 46 and 81 of human IGHV1-8-derived antibodies) that are highly mutated (see for instance DOI: 10.3389/fimmu.2017.01433), sites that may be missed by a nucleotide-focused prediction tools. The authors ought to discuss the structural context and its role in prediction. Are combinations of models as applied here better able to incorporate the structural perspective than without the need for explicitly incorporating a structural component into the computational process?

The authors removed sequences that had >10 mutations in any window of 20 consecutive sites (lines 442-445). I understand the rational but how does this impact in particular analysis of diversification of CDRs where higher levels of diverse substitutions may be observed. Please discuss.

Minor comments

Lines 338-346 appears to be Discussion and should be moved to that section.

The authors should indicate the reference sets (incl version number) that was used to annotate germline gene origin.

Have models been trained solely on sequences derived from the same germline gene or also on sequences derived from other genes? Residues far apart in the linear sequence may come into close proximity in the folded protein and this may impact mutability in particular of residues that are structurally important. Please comment.

Reviewer #2: Timely benchmark: EPAM (code released) shows nucleotide-context SHM models outperform protein LMs across human repertoires and a Replay system. With the revisions, this would be an important community reference.

**Have the authors made all data and (if applicable) computational code underlying the findings in their manuscript fully available?**

Reviewer #1: **No: **Raw data has been obtained from external sources. They, not the present authors, might not be able to release the data.

Reviewer #2: Yes

PLOS authors have the option to publish the peer review history of their article (what does this mean?). If published, this will include your full peer review and any attached files.

Reviewer #1: No

Reviewer #2: No

**Figure resubmission:**
---

## [Decision Letter · Decision Letter 1]

17 Nov 2025

Dear Dr. Matsen IV,

We are pleased to inform you that your manuscript 'Nucleotide context models outperform protein language models for predicting antibody affinity maturation' has been provisionally accepted for publication in PLOS Computational Biology.

Best regards,

Alexey Onufriev

Academic Editor

PLOS Computational Biology

James Faeder

Section Editor

PLOS Computational Biology

Reviewer's Responses to Questions

**Comments to the Authors:**

Reviewer #1: The comments made by the reviewers have been appropriately addressed.

Reviewer #2: All my comments have been addressed. I have no additional comments.

**Have the authors made all data and (if applicable) computational code underlying the findings in their manuscript fully available?**

Reviewer #1: Yes

Reviewer #2: None

PLOS authors have the option to publish the peer review history of their article (what does this mean?). If published, this will include your full peer review and any attached files.

Reviewer #1: No

Reviewer #2: No

---

## [Editor Report · Acceptance letter]

PCOMPBIOL-D-25-01510R1

Nucleotide context models outperform protein language models for predicting antibody affinity maturation

Dear Dr Matsen IV,

I am pleased to inform you that your manuscript has been formally accepted for publication in PLOS Computational Biology. Your manuscript is now with our production department and you will be notified of the publication date in due course.

With kind regards,

Zsofia Freund
